# Carbon Quantum Dots Modified (002) Oriented Bi_2_O_2_CO_3_ Composites with Enhanced Photocatalytic Removal of Toluene in Air

**DOI:** 10.3390/nano10091795

**Published:** 2020-09-09

**Authors:** Junping Ding, Huanchun Wang, Yidong Luo, Yushuai Xu, Jinsheng Liu, Ruichu Lin, Yuchen Gao, Yuanhua Lin

**Affiliations:** 1State Key Laboratory of New Ceramics and Fine Processing, School of Materials Science and Engineering, Tsinghua University, Beijing 100084, China; djp15@mails.tsinghua.edu.cn (J.D.); ydluozd@163.com (Y.L.); xuyushuai5736@163.com (Y.X.); 2China Astronaut Research and Training Center, Beijing 100094, China; goldsix@sohu.com (J.L.); linruichu@163.com (R.L.); accgao@163.com (Y.G.); 3High-Tech Institute of Xi’an, Xi’an 710025, China

**Keywords:** Bi_2_O_2_CO_3_, carbon quantum dots (CQDs), crystal orientation, toluene removal, photocatalysis

## Abstract

In work, (002) oriented flower-like Bi_2_O_2_CO_3_(BOC) composites are synthesized by a facile chemical route and carbon quantum dots (CQDs) are modified on their surfaces through a hydrothermal method. The synthesized samples (CQD/BOC) are characterized by X-ray diffraction (XRD), SEM, X-ray photoelectron spectroscopy (XPS), UV-Vis diffuser reflectances (DRS), BET and TEM/HRTEM. The morphologies of CQD/BOC composites are the flower-like shapes, the irregular flaky structures and the fine spherical particles of CQDs attached. Photocatalytic performances were investigated in terms of removing gaseous toluene at a concentration of 94.3ppm in air, with the assistance of CQD/BOC under artificial irradiation. Our results show that CQDs modified (002) oriented Bi_2_O_2_CO_3_ exhibits good photocatalytic activity for toluene decomposition, which can be attributed to the enhanced efficient charge separation. A certain ratio composite photocatalyst (BOC-CQD-15) shows a toluene removal rate of 96.62% in three hours, as well as great stability. CO_2_ was verified to be the primary product. The oriented flower-like Bi_2_O_2_CO_3_ with carbon quantum dots on the surface shows great potential in the field of solar driven air purification.

## 1. Introduction

Volatile organic compounds (VOCs) are one of the major gas pollutants in indoor air, including various alcohols, aromatics (benzene, ethylbenzene, toluene, xylene, etc.), aldehydes (acetaldehyde, formaldehyde, etc.) and halocarbons, which put humans health at great risks [1,2,3]. Many technologies are used to remove gas pollutants, such as adsorption by carbon-based filter media, ionization, ultraviolet degradation, plasma technology, catalytic degradation and the photocatalysis method [4,5,6]. Among all these technologies, photocatalysis may be a promising technique for indoor air purification, because harmless CO_2_ and H_2_O are the main products of pollutant degradation [7].

Bi-based photocatalysts have drawn increasing attention recently, because of their good chemical stability under visible light irradiation and unique electronic band structure. The hybridization of O 2p and Bi 6s orbitals in Bi-based photocatalysts composites results in a well-dispersed valence band (VB). In the meantime, the lone-pair distortion of Bi 6s orbitals can cause the pronounced overlap of O 2p and Bi 6s orbitals, which would increase the mobility of charge carriers and decrease the band gap [8,9,10]. Considering the stability of Bi^3+^, most studies of Bi-based photocatalysts have focused on Bi^3+^-containing compounds, such as Bi_2_O_3_, BiVO_4_, Bi_2_WO_6_, BiOX (X = Cl, Br, I), Bi_2_O_2_CO_3_ (BOC) and so on [11,12,13,14,15].

As a member of Bi-based photocatalysts, Bi_2_O_2_CO_3_ has been used for health care and in medical fields due to its no toxicity merit for a long time [16]. For first time, Bi_2_O_2_CO_3_ was found that could display good photocatalytic activity under UV light irradiation for the degradation of methyl orange aqueous solution in 2010 [17]. Bi_2_O_2_CO_3_ crystallizes in a body-centered orthorhombic Imm2 space group with lattice parameters of a = 3.865 Å, b = 3.862 Å and c = 13.675 Å [18]. Bi_2_O_2_CO_3_ has a featured layer structure, in which the CO_3_^2−^ layers are alternately intercalated between [Bi_2_O_2_]^2+^ polycation layers. The separate [Bi_2_O_2_]^2+^ and CO_3_^2−^ layers favor an internal electric field (IEF), which can greatly improve the charge separation rate [19]. Due to its particular layer structure, suitable band gap, excellent photocatalytic activity and long-term stability, Bi_2_O_2_CO_3_ has drawn considerable attention for its promising application in the photocatalysis field [20]. However, Bi_2_O_2_CO_3_ has a wide band gap (2.8–3.5) eV [21] and it can only response ultraviolet light of the solar light, which greatly restricts its practical application under full solar light. To solve this problem, several strategies have been investigated, mainly focusing on doping elements [22,23,24], Crystal facet engineering [18,25,26] and Constructing heterostructures [27,28,29]. As an up-conversion material, Carbon quantum dots (CQDs) can absorb NIR light at specific wavelengths and emit UV or visible light, which provide an indirect route for the use of NIR light in the photocatalytic process. Therefore, full-light-response photocatalytic heterostructures composed of CQDs and semiconductors, including CQDs/TiO_2_, CQDs/Cu_2_O, CQDs/Fe_2_O_3_ and CQDs/Ag_3_PO_4_, have recently been developed [30,31,32,33]. Thus, we were inspired that CQDs would extend the photo response range of Bi_2_O_2_CO_3_ through establishing a heterostructure.

Herein, we report a facile chemical method to fabricate Bi_2_O_2_CO_3_ composites with (002) facet exposed and a hydrothermal route to modified carbon quantum dots on the surface of Bi_2_O_2_CO_3_. The morphologies of synthesized composites (CQD/BOC) were flower-like shapes, irregular flaky structures and fine spherical particles. Their photocatalytic properties for removal toluene in air under the artificial irradiation are investigated comparatively. Based on the close correlation between the structure characteristics and physicochemical properties of the material, BOC-CQD-15 has proved to be most active, with a removal rate of up to 96.62%. Permineralization of toluene in photocatalysis is proposed based on the characterization; CO_2_ was verified to be the primary product. This work may probably extend to the application of oriented flower-like Bi_2_O_2_CO_3_ with carbon quantum dots on the surface in air purification.

## 2. Materials and Methods

### 2.1. Fabrication of CQD/BOC Membranes

All the chemical reagents in this work were analytical-grade and used without any further purification.

#### 2.1.1. Preparation of Bi_2_O_2_CO_3_ Composites

An amount of Bi_2_O_3_ (99.99%, Aladdin Industrial Corporation, Shanghai, China) was dissolved in 1mol/L HNO_3_ (GR, Sinopharm Chemical Reagent Co., Ltd., Shanghai, China) solution through the ultrasonic sound. The solution was put in a 20 °C water bath until the equalization of temperature. Then, 0.6mol/L Na_2_CO_3_ (RA, Sinopharm Chemical Reagent Co., Ltd., Shanghai, China) solution was added to the above solution at a speed of 30mL/min, so as to obtain a pH = 7 uniform solution, while stirring manually. Finally, the product was centrifuged, washed with deionized water and ethyl alcohol and dried at 70 °C in an oven to obtain Bi_2_O_2_CO_3_ composites labeled as BOC.

#### 2.1.2. Preparation of CQDs

The CQDs were obtained through a hydrothermal route [34]. Glucose (1 g, AR, Sinopharm Chemical Reagent Co., Ltd., Beijing, China) was dissolved into deionized water (80 mL) to obtain a homogeneous solution. Then, the solution was treated under a hydrothermal condition (180 °C, 4 h). After that, the solution was given a filter treatment (0.1 μm, nylon), and then a reddish-brown CQDs suspension was obtained.

#### 2.1.3. Preparation of CQDs/BOC Composites

A total of 0.25 g BOC was added to 70 mL deionized water with ultrasonic dispersant for 15 min. Subsequently, a certain volume of CQDs suspension was dropped into above mixture. Then, the mixtures were sealed in a Teflon-lined stainless-steel autoclave and heated at the temperature 180 °C under autogenous pressure for 4h. After natural cooling to room temperature, the product was filter centrifuged, washed with deionized water and ethyl alcohol and dried at 70 °C in an oven to obtain CQD/BOC composites. To study the impact of the CQDs content on the photocatalytic performance of the composites, a series of the CQD/BOC composites were obtained by adjusting the volumes of CQDs suspension (5, 10, 15 and 20 mL). The specimens were correspondingly labeled as BOC-CQD-5, BOC-CQD-10, BOC-CQD-15, BOC-CQD-20.

#### 2.1.4. Fabrication of CQD/BOC Membranes

A total of 0.025 g BOC or CQD/BOC was added to 150 mL deionized water with ultrasonic dispersant for 15 min. The mixtures were filtered at 0.1 μm nylon membrane by vacuum pump. The BOC or CQD/BOC photocatalysic thin-film on the nylon membrane was obtained. The CQD/BOC membranes were fabricated, as illustrated in Figure 1.

### 2.2. Characterization

The powder X-ray diffraction (XRD) patterns were obtained from a diffractometer (D8-Advance, Bruker, Karlsruhe, Germany) using monochromatized Cu Kα (λ = 1.54056 nm) radiation with scanning speed of 0.15°/s. The morphology of the samples was carried out on a scanning electron microscope (JSM-7001F, JEOL, Tokyo, Japan) operating at a 5 kV and a high-resolution transmission emission electron microscope (JEM-2100F, JEOL, Tokyo, Japan). The XPS spectra measurements were conducted on an X-ray photoelectron spectroscopy (ESCALAB 250Xi, Thermo Fisher, California, USA). The specific surface area was measured on an automated gas sorption analyzer (AutosorbiQ2, Quantachrome, Florida, USA). UV-Vis diffuser reflectances (DRS) were carried out on a UV-Vis spectrometer (Lambda 950, PerkinElmer, Massachusetts, USA).

### 2.3. Photocatalytictest

The photocatalytic properties of the specimen were evaluated by toluene removal in air using a gas phase photocatalysis system (FPCS-1, Beijing Ferren Science & Technology Co., Ltd., Beijing, China). Before the photocatalytic test, the inter space of the reactor was first substituted with nitrogen to expel the oxygen and moisture. Toluene standard gas with concentration of 94.3 ppm in air was used as reactant. The BOC or CQD/BOC membrane was placed at the bottom of the reactor. Toluene standard gas was pumped into the chamber (about 450 mL). Then, the reactor was kept in dark for 30 min to reach the adsorption equilibrium. An incident light source (a 300W xenon lamp) was placed above the reactor which has a quartzose cover as an upper surface. At regular time intervals, the mixture gas in reactor was analyzed by gas chromatograph equipped with two flame-ionization detectors (FID). Toluene analysis was carried out with FID loaded with an Rt-Q-Bond Plot column (30 m × 0.25 mm, film thickness 10 μm), while CO_2_ analysis was carried out with the other FID loaded with a packed column (TDX-01, 3 m × 3 mm), followed by a methanizer CO_2_ concentration. The gas samples were fed to GC online through an automatic gas sampling valve. The temperature of the reactor was controlled using circulating cooling water to avoid thermal effect during the degradation process.

## 3. Results and Discussion

### 3.1. Morphology Analysis

The morphologies of the as-prepared BOC and CQD/BOC samples were observed by SEM and TEM. It is clear that the morphologies of the pure BOC were flower-like shape with a diameter of ca. 6um (Figure 2a), and the irregular flaky structures of BOC nanosheets gathered together with a diameter of ca. 500 nm (Appendix A). The morphologies of CQD/BOC were flower-like shapes (Figure 2c–f), irregular flaky structures (Figure 2b and Appendix A) and fine spherical particles (Figure 2d and Appendix A). Thus, more reactive sites could be provided, due to the higher surface-to-volume ratio of BOC and CQD/BOC.

The BET surface area test results of as-prepared samples are shown in Table 1. It was found that the specific surface area and pore volume of the sample by chemical route were much larger than the BOC by a hydrothermal method [35]. The specific surface area and pore volume of CQD/BOC were about 50% larger than BOC. They are similar to flower-like Bi_2_O_2_CO_3_ [34].Therefore, the larger BET surface area of CQD/BOC may result in better photocatalytic performance by providing more actives sites than BOC sample.

The morphology of BOC-CQD-15 was also observed by TEM and HRTEM (Figure 3 and Appendix A). As shown in Appendix A, the main morphology of sample was flower-like shape. Simultaneously, there are some fine spherical particles and irregular flaky structures, which is in accordance with SEM results. To further elucidate the element distribution of BOC-CQD-15, energy-dispersive X-ray elemental mappings were employed (Figure 3b–d), where Bi, O and C elements were uniformly distributed in the obtained sample. These mapping images correspond to the TEM image shown in Figure 3a. To determine the C element content, the EDS spectrums of the spectrum 4 and 5 in Figure 3g were measured. The C element content of spectrum 4 was 78.68%, which was much larger than 55.38% of spectrum 5. It revealed that the deep color dots were carbon quantum dots with the diameter of ca. 5–30 nm. Additionally, in Figure 3e,f, the (013) crystalline of BOC could be found in BOC-CQD-15 according to the lattice spacing of 0.291 nm (JCPDS 41-1488). Furthermore, a lattice spacing of 0.320 nm could also prove that the introduction of CQDs (004) according to JCPDS 26-1080. The result further indicates that CQDs were successfully modified on the surface of Bi_2_O_2_CO_3_.

### 3.2. Structure and Composition Analysis

The phase structure of the obtained BOC and CQD/BOC samples were detected by X-ray powder diffraction (XRD), and the results are shown in Figure 4. It reveals that all diffraction peaks of different samples could be well indexed to the pure phase of Bi_2_O_2_CO_3_ (JCPDS 41-1488), without impurity peaks. The diffracted intensity ratio of (002)/(013) in the BOC, BOC-CQD-5, BOC-CQD-10, BOC-CQD-15 and BOC-CQD-20 were 59.50%, 65.17%, 52.74%, 76.95% and 83.55%, respectively, which were much larger than 25% of the primitive BOC standard card (JCPDS 41-1488). (002) facet was exposed dominantly, which might contribute to the separation of photo-excited hole-electron pairs [36].

X-ray photoelectron spectroscopy (XPS) was conducted to investigate the chemical composition and surface electron state of the CQD/BOC samples (Figure 5 and Appendix A). Figure 5a demonstrated the typical survey spectrum of the as-obtained samples, showing that all of the samples consisted solely of Bi, O and C. The high-resolution XPS spectra of C1s, O1s and Bi4f for the obtained photocatalysts are shown in Figure 5b−f. In Figure 5b, the C1s peak at a binding energy of 284.8 eV can be attributed to the C-C bond with sp^2^ orbital; the peak observed at 289.0 eV should be ascribed to the C–O bond in Bi_2_O_2_CO_3_ [37]. The spectra of O1s can be fitted into three Gaussian-Lorenzian peaks (see Figure 5c and Appendix A). The peak located at 529.8 eV is ascribed to the lattice oxygen in Bi−O binding energy, and the peaks at 530.7 eV and 531.6 eV can be assigned to carbonate and the surface hydroxyl groups on the surface of Bi_2_O_2_CO_3_ [14,38]. In Figure 5d–f, the two apparent characteristic peaks for Bi-4f located at 159.1eV and 164.4 eV are attributed to Bi-4f_7/2_ and Bi4-f_5/2_ in Bi_2_O_2_CO_3_, indicating the existence of Bi^3+^ in the samples. The Bi-4f_7/2_ and Bi-4f_5/2_ of CQD/BOC have a negative shift to low binding energy compared with BOC, indicating the higher electron density around Bi elements in CQD/BOC samples, and proving CQDs modified on the surface of Bi_2_O_2_CO_3_ [39,40].

### 3.3. Photocatalytic Properties

The photocatalytic properties of the as-prepared samples were investigated through removing gaseous toluene (94.3 ppm) in air under the irradiation of an incident light source. As shown in Figure 6a, the pure BOC achieved a good photocatalytic performance with a removal rate of 70%, which can be attributed to (002) crystal face exposed. However, CQD/BOCs were superior to the pure BOC, with a removal rate of up to 95%, which can be attributed to CQDs and (002) crystal face. The photocatalytic property of BOC-CQD-15 reached 96.62% after three hours irradiation.

In order to get a deeper understanding of the degradation reaction of toluene in air, the degradation products were detected using gas chromatograph (GC) with flame-ionization detectors (FID). The results are shown in Figure 6b and Appendix A. The CO_2_ productivity of BOC-CQD-15 was 38.5 μmol, which is 2.4 times of the pure BOC and was the highest in all the CQD/BOCs.

On the basis of the toluene removal rate and the CO_2_ productivity, the photocatalytic property of BOC-CQD-15 was the best in all the CQD/BOCs.

In order to observe the photo response range of CQD/BOC, the phototcatalytic properties of BOC-CQD-15 were investigated through its ability of toluene decomposition under the irradiation of infrared light, visible light and ultraviolet light (showed in Figure 6c). BOC-CQD-15 has not response to IR light and Vis light, but it has response to UV light.

The stability of photocatalytic degradation of toluene in air was observed by repeating the experiment for five runs under full spectrum after ultraviolet irradiation, and the result is shown in Figure 6d and Appendix A. It was clear that the photocatalytic property of BOC-CQD-15 under full spectrum was superior to its property under ultraviolet due to the outstanding up-converted photoluminescence peculiarity of carbon quantum dots modified on the surface of Bi_2_O_2_CO_3_, which further extend the photoresponse range of BOC to the near infrared light. The BOC-CQD-15 could remain a constant photocatalytic performance as high as 90% in terms of removing toluene under incident light irradiation after five recycling runs. This phenomenon revealed a good recyclability of BOC-CQD-15 for toluene removal.

### 3.4. Photocatalytic Degradation Mechanism

The UV-Vis diffuse reflectance spectra of the samples were examined, and the results presented in Figure 7. It can be seen that the bandgap of all the samples can be decided about (3.4~3.5) eV. The results reveal that the CQD scarcely influence the light absorption of BOC or change the band gap of samples obviously.

According to the characterization of chromatogram, CO_2_ is the primary product, and there is a little of CO during toluene degradation reaction. The CO productivity of BOC-CQD-15 was 6.33 μmol, which was 2.5 times of the pure BOC (showed in Appendix A). No other byproducts or intermediate were detected. It indicated that the main products of toluene degradation were CO_2_ and H_2_O and there was a modicum of CO.

The photocatalytic degradation mechanism of VOCs in air is slightly different from that in aqueous solution. A possible photocatalytic mechanism of the CQD/BOC composites toward the removal of toluene under simulated sunlight irradiation is schematically depicted in Figure 8. The BOC can effectively respond to the light with the wavelength shorter than about 400 nm. When the CQD/BOC photocatalyst reacts to the photons, the electrons are excited from the valence band to the conduction band of BOC, thus producing electron-hole pairs. Simultaneously, it is generally accepted that carbon quantum dots are an outstanding up-converted photoluminescence material. The up-converted emissions are usually located at shorter wavelengths in the range of 300–650nm [38]. As a result, a part of the up-converted emissions of CQDs can in turn excite BOC to generate additional photo excited charges, further extending the photoresponse range of BOC to the NIR light. Meanwhile, CQDs can also be excited by absorbing visible light, the π electrons or σ electrons are excited to the lowest unoccupied molecular orbital (LUMO) [41,42]. The excited CQDs can act as excellent electron donors and acceptors. Consequently, the CB electrons in BOC would transfer to CQDs (π or σ orbital), which help with the separation and the migration of photo excited carriers.

The photogenerated electrons which migrate to the surface of BOC and CQDs reduce the surface-adsorbed O_2_ to highly active species •O_2_^−^. Thus, highly active •O_2_^−^ oxidizes toluene to CO_2_, H_2_O and other intermediate products, such as benzene, benzoic acid, benzaldehyde and benzyl alcohol [43]. Then h^+^ on the surface of BOC oxidizes the surface-adsorbed H_2_O to highly active species •OH, and the active species (•O_2_^−^ and •OH) oxidizes the adsorbed intermediates to CO_2_ and H_2_O, consequently forming the final products CO_2_ and H_2_O.

## 4. Conclusions

In summary, (002) oriented flower-like Bi_2_O_2_CO_3_ composites were synthesized by a facile chemical route and carbon quantum dots were modified on their surface through a hydrothermal method. The synthesized composites (CQD/BOC) have three morphologies, which were flower-like shapes, irregular flaky structures and fine spherical particles. Through HRTEM characterization, it was proved that CQDs were modified successfully on the surface of Bi_2_O_2_CO_3_. Photocatalytic mineralization of toluene in air over CQD/BOC was measured. The effect of BOC-CQD-15 was optimum, and as a result, the toluene removal rate was up to 96.62% in three hours under full light irradiation, the rate was still up to 90% after five recycling runs in terms of stability. CO_2_ was verified to be the main product after reaction. Better performance can be contributed to (0 0 2) facet orientation evolution and CQDs modified on the surface of Bi_2_O_2_CO_3_, which enhance the efficient separation of photogenerated electron-holes.

## Figures and Tables

**Figure 1 nanomaterials-10-01795-f001:**
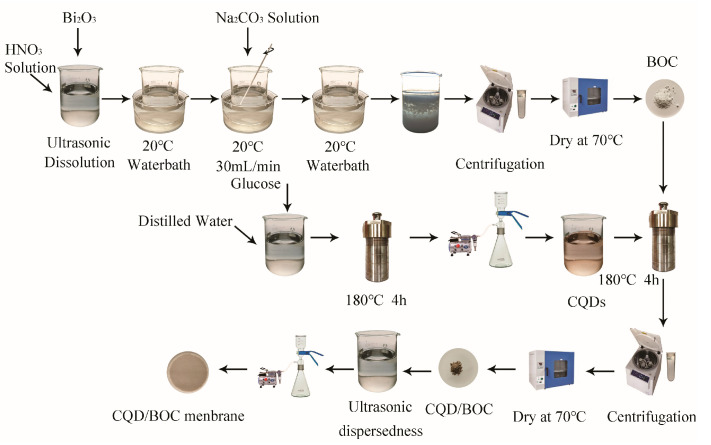
Chemical route for the preparation of carbon quantum dots (CQD)/Bi_2_O_2_CO_3_ (BOC) membrane.

**Figure 2 nanomaterials-10-01795-f002:**
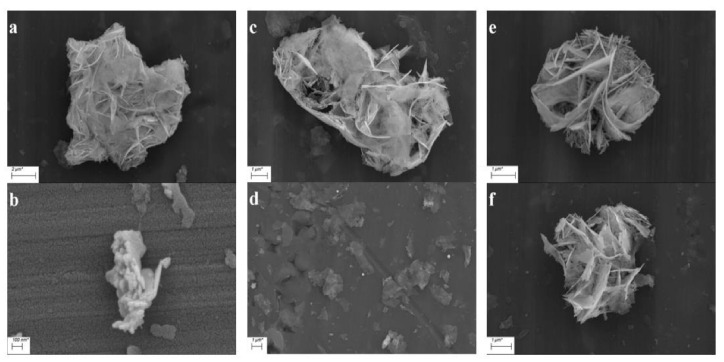
SEM images of BOC (**a**), BOC-CQD-5 (**b**), BOC-CQD-10 (**c**,**d**), BOC-CQD-15 (**e**) and BOC-CQD-20 (**f**).

**Figure 3 nanomaterials-10-01795-f003:**
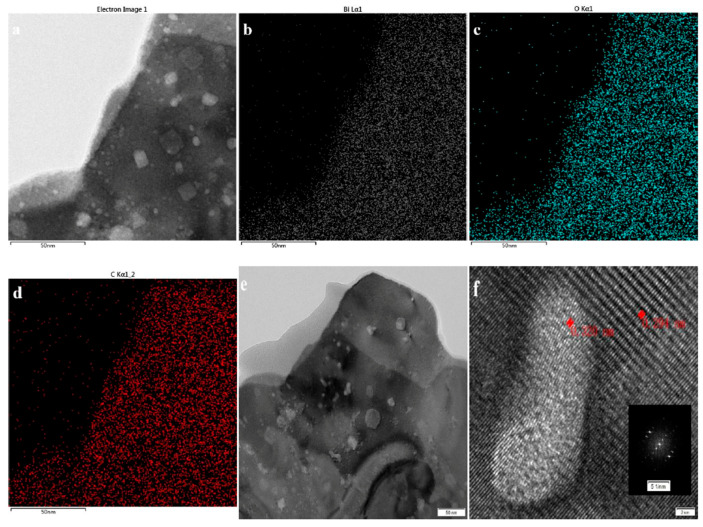
Element mapping images (**a**–**d**), HRTEM images (**e**,**f**) and EDS spectrum (**g**) of BOC-CQD-15.

**Figure 4 nanomaterials-10-01795-f004:**
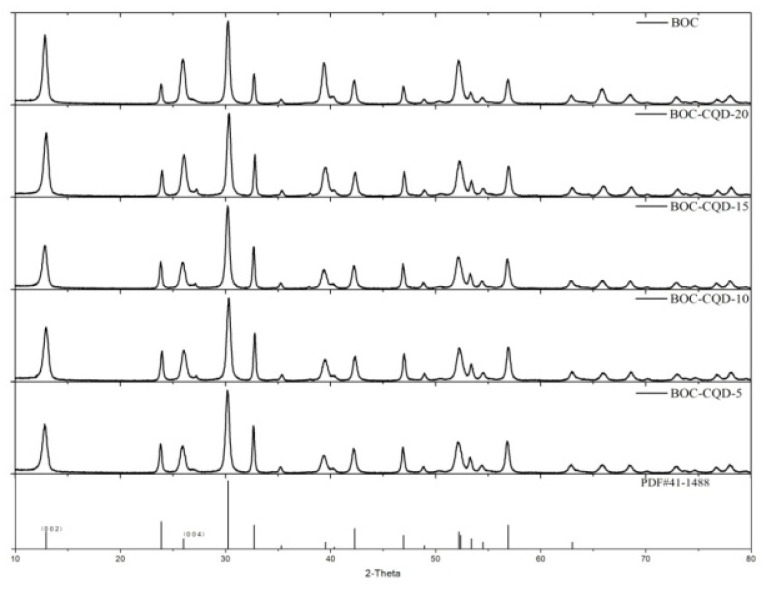
X-ray diffraction (XRD) patterns of as-prepared samples.

**Figure 5 nanomaterials-10-01795-f005:**
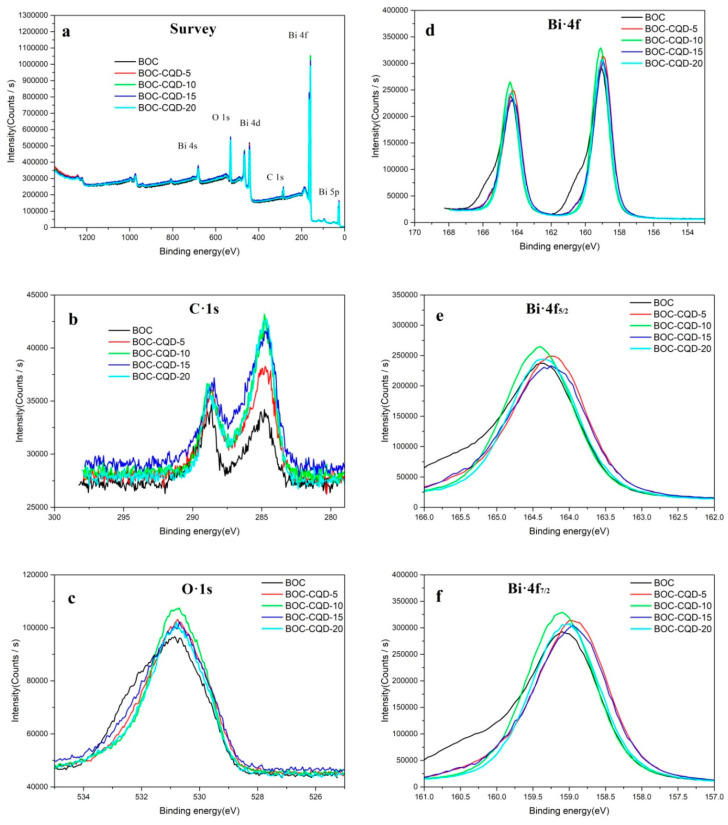
X-ray photoelectron spectroscopy (XPS) spectra of BOC samples: survey scan (**a**), C1s (**b**), O1s (**c**), and Bi-4f (**d**–**f**).

**Figure 6 nanomaterials-10-01795-f006:**
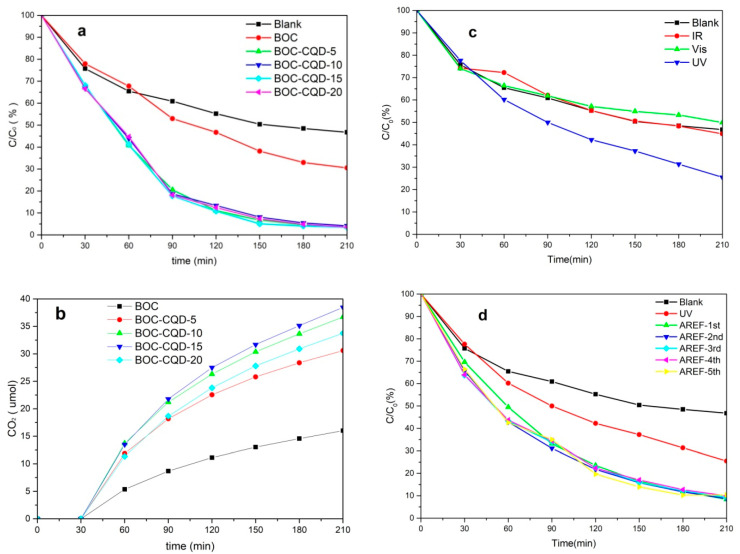
The photocatalytic properties of the as-prepared samples for toluene removal in air (**a**), the CO_2_ productivity of the samples (**b**), the ability of BOC-CQD-15 under IR, Vis, UV irradiation (**c**) and the stability of BOC-CQD-15under full light during five cycles (**d**).

**Figure 7 nanomaterials-10-01795-f007:**
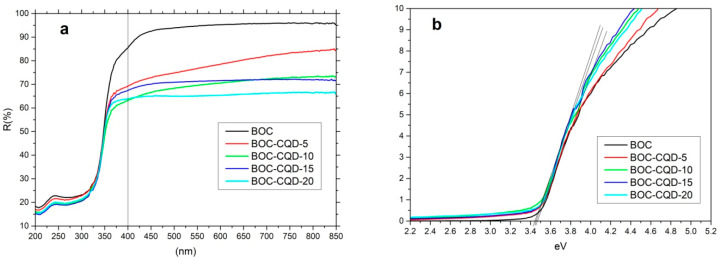
UV-Vis diffuse reflectance spectra of as-prepared samples (**a**) and band gap fitting with KMrelation (**b**).

**Figure 8 nanomaterials-10-01795-f008:**
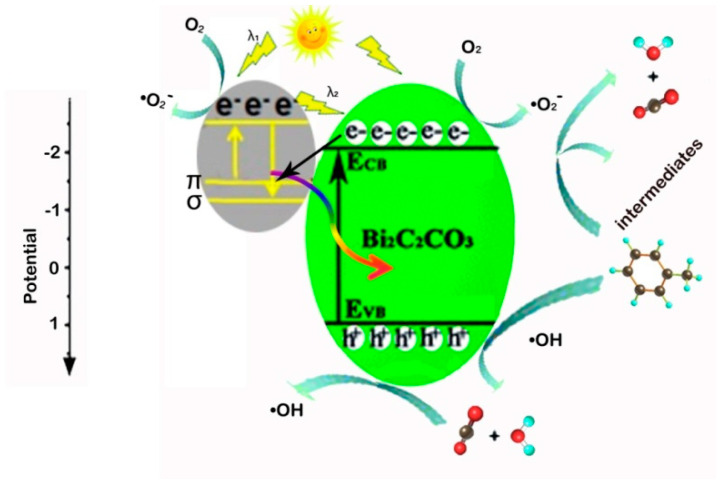
Photocatalytic mechanism of toluene removal for BOC-CQD-15.

**Table 1 nanomaterials-10-01795-t001:** Specific surface area of BOC and CQD/BOC.

	BOC	BOC-CQD-5	BOC-CQD-10	BOC-CQD-15	BOC-CQD-20
Surface area (m^2^/g)	12.293	17.189	18.842	17.699	17.266
Total pore volume for pores with Diameter less than 194.68 nm at P/Po = 0.990027 (cm^3^/g)	0.0952	0.1441	0.1254	0.1248	0.0934

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
