# Peer review of "Carbon Quantum Dots Modified (002) Oriented Bi_2_O_2_CO_3_ Composites with Enhanced Photocatalytic Removal of Toluene in Air"

_nanomaterials, 2020, doi:10.3390/nano10091795_

Round 1

Reviewer 1 Report

In this work, the author synthesized CQD/BOC composites for the removal of toluene in air. The reviewer finds that the manuscript is overall well organized and publishable after revision. Comments to this manuscript are:

  1. The authors compared the photocatalytic properties of CQD/BOC at various conditions in Figure 5. There are several lines disconnected between dots. It seems that the authors may delete some data points which exist far from the trend lines. For better understanding, the authors should update this. The authors may provide the error bars based on the statistical analysis on some important data sets.
  2. The authors used a Xenon lamp to irradiate the CQD. The photocatalytic properties increased with the addition of CQD may depend on the light absorption of the CQD. The extinction profile of the CQD and the wavelength dependence of CQD used in the photocatalytic measurements need be discussed.

Author Response

Dear reviewer,

Thanks very much for giving us further opportunity to improve our manuscript. We have conducted a major revision, hoping that by the newly revised and modified version of the paper your prestigious journal’s standard can be satisfied. These changes and the responses to respectable reviewers have been listed below.

(1) The authors compared the photocatalytic properties of CQD/BOC at various conditions in Figure 5. There are several lines disconnected between dots. It seems that the authors may delete some data points which exist far from the trend lines. For better understanding, the authors should update this. The authors may provide the error bars based on the statistical analysis on some important data sets.

Reply: The reviewer's view is quite correct. However, the disconnected between dots in previous version were caused by the missing of several data points due to mis-operation rather than deletion intentionally. As the reviewer’s suggestion, we have redrawn Figure 5 and updated it (Please see page 8 Figure 6). The interval was set to 30 min in order to show the variation tendency of C/C0 along with irradiation time based on the previous experimental results. We hope the redrawn figure will beneficial for better understanding the photocatalysis property of BOC and BOC-CQDs.

(2) The authors used a Xenon lamp to irradiate the CQD. The photocatalytic properties increased with the addition of CQD may depend on the light absorption of the CQD. The extinction profile of the CQD and the wavelength dependence of CQD used in the photocatalytic measurements need be discussed.

Reply: As the reviewer’s suggestion, we have carried outUV-Vis-NIR diffuser reflectances (DRS) of all the samples (Please see page 4 line 121 ~ 122 and page 9 Figure 7). As expected, the results reveal that the CQD scarcely influence the light absorption of BOC or change the band gap of samples obviously. (Please see page 9 line 231~234).

We hope the revised manuscript is now appropriate for the publication in the journal Nanomaterials.

Reviewer 2 Report

Your manuscript entitled “Carbon quantum dots modified (0 0 2) oriented Bi2O2CO3 composites with enhanced photocatalytic removal of toluene in air” is interesting for Nanomaterials. It shows that Carbon quantum dots (CQDs) are modified on the surface of Bi2O2CO3 (BOC). The morphologies are flower-like shapes, irregular flaky structures, and fine spherical particles. The flower-like shapes (002) oriented BOC/CQDs exhibits the good photocatalytic activity for toluene removal. However, I think that the manuscript needs to be rearranged (major revision) prior the acceptance in Nanomaterials. I honestly have difficulties to follow the story of your manuscript. The suggestions are following:

  1. Sample preparations at the end? how you control all compositions in those CQD? You should explain how to make BOC and BOC/CODs with different ratios before “2.1. Structure composition and analysis” (page 2, line 78).
  2. After sample preparations, one should first analyse SEM (page 5, line 116). The first sample appearance is more important than the structure composition analysis.
  3. The photocatalytic properties in Fig. 5 are not clear. So, how you determine that BOC-CQD-15 should be explored more for the stability as you wrote on page 7, line 144. For me, the photocatalytic properties reached 98% for BOC-CQD-15 is not a good excuse without the comparison to others. Some lines in Figure 5a, 5b, and 5c are broken so it is confusing whether they are the same series or not.

Author Response

Dear reviewer,

We are so thankful for considering our manuscript to review and giving us further opportunity to improve it. We tried to change the paper according to what we were asked to do. These changes and the responses to respectable reviewers have been listed below. We hope our revision could improve the paper to a level of your satisfaction.

(1) Sample preparations at the end? how you control all compositions in those CQD? You should explain how to make BOC and BOC/CODs with different ratios before “2.1. Structure composition and analysis” (page 2, line 78).

Reply: As the reviewer’s suggestion, we have rearranged our paper. The  Materials and Method section is present as the second section followed by the Result and Discussion section for Better understanding of our work. Please see section 2 Materials and Methods on Page 2 and section 3 Results and Discussion on Page 4. By adjusting the volumes of CQDs suspension (5, 10, 15 and 20 mL) during preparation of CQD/BOC composites, a series of samples with various CQDs content were obtained (Please see page 3 line 102 ~ 104).

(2) After sample preparations, one should first analyse SEM (page 5, line 116). The first sample appearance is more important than the structure composition analysis.

Reply: As the reviewer’s suggestion, we have rearranged our article. (Please see line 140 ~ 201)

(3) The photocatalytic properties in Fig. 5 are not clear. So, how you determine that BOC-CQD-15 should be explored more for the stability as you wrote on page 7, line 144. For me, the photocatalytic properties reached 98% for BOC-CQD-15 is not a good excuse without the comparison to others. Some lines in Figure 5a, 5b, and 5c are broken so it is confusing whether they are the same series or not.

Reply: The approach of the respectable reviewer is quite correct and logical. As the reviewer’s suggestion, we have redrawn Figure 5 and the interval was set to 30 min based on the previous experimental results (Please see page 8 Figure 6). The variation tendency lines of C/C0 along with irradiation time are more legible. For better present our work, Figure 6 was rearranged. The discussion was carried out more rigorous. (Please see Page 8 line 203 ~214).

We hope the revised manuscript is now appropriate for the publication in the journal Nanomaterials.

Round 2

Reviewer 2 Report

Thanks for your amendments. It is OK for the publication in Nanomaterials.